# Total Life Cycle of Polypropylene Products: Reducing Environmental Impacts in the Manufacturing Phase

**DOI:** 10.3390/polym12091901

**Published:** 2020-08-24

**Authors:** Viktoria Mannheim, Zoltan Simenfalvi

**Affiliations:** 1Higher Education Industrial Cooperation Center, University of Miskolc, 3515 Miskolc-Egyetemváros, Hungary; 2Department of Chemical Machinery, Faculty of Mechanical Engineering and Informatics, University of Miskolc, 3515 Miskolc-Egyetemváros, Hungary; simenfalvi@uni-miskolc.hu

**Keywords:** polypropylene products, complete life cycle, life cycle assessment, looping method, injection-moulding process

## Abstract

This paper assesses the environmental burdens of a polypropylene product throughout the product’s life cycle, especially focusing on the injection-moulding stage. The complete life cycle model of the polypropylene product has been developed from the raw material extraction and production phase through its usage to the end-of-life stage with the help of the life cycle assessment method. To find the answers to the posed problems, different impacts were analysed by GaBi 8.0 software. The analysis lasted from the cradle to the grave, expanding the analysis of the looping method. The aim of the research was to determine the energy and material resources, emissions, and environmental impact indicators. Basically, the article tried to answer three questions: (1) How can we optimize the production phase for the looping method? (2) Which materials and streams are recyclable in the design of the production process? (3) What is the relationship between life cycle stages and total life cycle of the product? As we inspect the life cycle of the product, the load on the environment was distributed as follows: 91% in the production phase, 3% in the use phase, and 6% in the end-of-life phase. The results of the research can be used to develop technologies, especially the injection-moulding process, with a lower environmental impact.

## 1. Introduction

The use of polymers in the technological sphere has recently received increased attention, and the utilization of injection-moulding processes is widespread in product manufacturing [1]. Nowadays, polypropylene is one of the most popular polypropylenes. Typical products made from moulded polypropylenes include storage containers, food packaging, crates, and parts for the automotive and vehicle industries. Requirements for manufacturing process optimisation in terms of environmental impact reduction and the reliability and safety of the structural elements have grown significantly [2].

Due to the importance of injection-moulding processes, research is increasingly concerned with life cycle assessment of moulded products. Life cycle assessment is the preferred method to analyse the environmental parameters of a polypropylene product. The life cycle approaches and methodologies can be applied to several aspects. The idea of a complete life cycle model in this research topic has already been raised by Civancik-Uslu et al. [3].

According to the European Environment Agency, the practical solutions aimed at a circular economy include eco-design, waste prevention programs, and extending the lifetime of products [4]. So, in the context of the circular economy and sustainable production, it is essential to evaluate the complete life cycle of polypropylene products. Focusing on safety within the circular economy can add a new level to life cycle assessment and may identify additional goals related to reduced environmental burdens and the protection of human health. Many studies have determined that the circular economy promotes closing loops in industrial systems, minimizing waste, and reducing raw material and energy inputs [4,5].

According to the conclusions of Szita [6], the prerequisite for sustainable life cycle management is an understanding of the various life cycle stages. During the innovative technological developments based on life cycle assessments, the production stage must be considered in particular, with a focus on the life cycle of products in the study by Labuschagne et al. [7].

A life cycle model with a looping method in the production phase promotes sustainability by maintaining the value of polypropylene products and minimizing resources and wastes [8]. Polypropylene scrap looping can become a good strategy in the production stage within the new framework defined by the circular economy. The use of renewable raw materials in the production stage has steadily increased over the last few years [9]. Grosso et al. [10] came up with a beneficial solution involving the use of recycled scrap instead of virgin material. Analysis of the complete life cycle of polypropylene products with a looping method within the European Union has not been previously performed. The reduction principle targets the minimization of raw material use, energy input, and waste production in the manufacturing phase, whereas the reuse principle refers to the repeated use of products for their intended purpose by Ghisellini et al. [11].

At the end-of-life (EoL) stage, we need to consider that most polypropylene products are not biodegradable. Polypropylene waste can be treated via disposal, incineration, or recycling processes. Recycling is mostly achieved by mechanical processes. The quality of the recycled polypropylene product mainly depends on the physicochemical properties of the polypropylene, as well as the processing conditions and the purity of the input polypropylene waste. Many studies have addressed these economic and ecological issues by limiting the large-scale application of chemical recycling processes [12,13]. The recycling of polyolefins requires advanced assessment protocols because the diffusion of a given substance is faster than in PET, with increasing migration rates and increasing sorption of contaminants [14,15]. Coulier et al. [16] showed increasing migration rates during repeated recycling of some polypropylene additives [16].

In the last 10 years, European Food Safety Authority (EFSA) published about 100 scientific opinions on recycled polypropylene and 5% of the processes described the recycling of polyolefins [17]. According to life cycle assessment research works [18,19], recycling polypropylene waste is generally environmentally preferred compared to incineration and landfill. Various studies [20,21,22,23] report that 22–43% of polypropylene waste is disposed of in landfills.

The environmental product declaration (EPD) is based on the life cycle assessment results and the product life cycle assessment is based on the relevant environmental product declaration modules. The EPDs for polypropylenes are based on life cycle inventory data from Polypropylenes Europe, according to ISO 14025 [24], and to the Polypropylenes Europe’s product category rules [25], which describe the production of the polypropylene polymer.

The aim of this research was to analyse the environmental burdens of a polypropylene product throughout the product’s life cycle, focusing on the injection-moulding stage. The main objective was to obtain information on the environmental impact associated with the injection moulding of polypropylene products. Three alternative production solutions were examined: (1) without looping method, (2) with process water looping only, and (3) with recirculated polypropylene scrap and water looping.

Furthermore, the goal of this research work was to propose an injection-moulding technology that can provide environmentally friendly engineering solutions. The first section of this paper presents the methodology, including the goal, scenarios, the determination of the functional unit and the system boundaries, the allocation method, and the applied software. The next section gives a description of the life cycle inventory (LCI) methodology and explains the life cycle impact assessment (LCIA) method.

This paper introduces and investigates the selected environmental impact categories, the material and energy resources, and the emissions data. The main section explains the obtained research results and the looping method; the last section provides the conclusions of the research work. The results can be used to design manufacturing processes involving a lower environmental impact and to improve the injection-moulding process’s environmental performance. This research work is relevant because the economization of resources is important in a life cycle assessment.

## 2. Materials and Methods

### 2.1. Methodology and Scenarios

The life cycle assessment methodology was employed in accordance with the recommendations provided by the ISO 14040 and 14044 standards [26,27]. This approach enables the analysis of the environmental impacts associated with stages in the life cycle of a polypropylene product, from the extraction of raw materials for their injection moulding until the used product becomes polypropylene waste. This work comprises the life cycle assessment phases, a life cycle inventory (LCI) analysis, a life cycle impact assessment (LCIA), and the interpretation of the research results.

This research work aims to obtain new information on the objective environmental impacts associated with polypropylene products in the European Union by comparing three stages: the production stage with injection-moulding process, the use stage with washing process, and the end-of-life stage with waste treatment. Polypropylene granules are produced in the European Union and processed in the local injection-moulding plant of polypropylene products. The granules are moulded and the products are washed in the use stage. The polypropylene products are managed as polypropylene waste in an incineration plant at the end-of-life stage.

This research work analyses three scenarios regarding the environmental impacts that define the life cycle of the polypropylene product. Different methodologies were established to distribute the material and energy consumption, as well as the emissions and waste generated, between the three different scenarios of the production stage. In the first scenario, the injection-moulding process is determined without looping. In this case, polypropylene scraps are treated, and the water from the cooling process is managed as wastewater in a wastewater treatment plant. The second scenario determines the injection moulding with water looping in the production stage. The modification of two looped parameters (polypropylene scrap and process water) was also analysed to respond to possible changes in the production stage, to identify which of the three scenarios is preferable from an environmental point of view (emissions and resources). Replacement and recycling were not calculated. The input‒output data for the production, use, and EoL stages of the third scenario were calculated, considering the reference flows.

### 2.2. System Boundaries and Functional Unit

The system boundaries were developed cradle-to-grave. The analysis begins with the production stage and these datasets are linked with use and end-of-life data to create complete life cycle inventories for the polypropylene product. This research contemplates the total life cycle, considering the stages of extracting the raw materials for injection moulding, the using of the polypropylene product with transport and washing, and the end-of-life stage thermic processing as polypropylene waste. Auxiliary systems such as transporting materials for use, obtaining electric power from a Hungarian energy mix, using diesel oil for transport of the product, and polypropylene waste are included in this research analysis. Equipment, machinery, and trucks are placed beyond the limits of the system—these are not relevant.

Polypropylene products have a 20-year lifetime and the analysed amount is moulded in 1 h. There is no rotation, so during this time they have one injection-moulding process in the production phase and one washing process in the use phase. In the end-of-life stage, the polypropylene products are incinerated as polypropylene waste and the energy is recovered.

First, the functional unit was defined as the distribution of 25 kg of polypropylene product output for the injection-moulding, use, and end-of-life stages. Considering the effect of the total life cycle of the polypropylene product, we redefined the functional unit as the distribution of 1 kg of product output.

### 2.3. Allocation

In the life cycle of the product, all materials and energy consumed, and process emissions, are associated with the polypropylene product that comes out of the injection-moulding process. In addition to the main product, this process produces polypropylene waste and wastewater. The allocation hierarchy is suggested by ISO 14044 [27]. The production process has been assigned as a function of the mass of the moulded polypropylene. For the transport of refinery products (diesel oils), the emission allocation has been based on mass. The energy demand has been assigned as a function of the energetic content.

### 2.4. LCA Software

The goal of this research work was to determine the material and energy resources, emissions, and environmental impact categories for the complete life cycle of a polypropylene product using a professional and extension dataset. After quantifying the environmental impacts, the life cycle analysis of the investigated system was carried out by applying GaBi 8.0 thinkstep software (by Sphera Solutions Ltd., Stuttgart, Germany). Normalized and weighted values of the life cycle stages were determined by the Higher Education Industrial Cooperation Centre (HEICC) of the University of Miskolc, Miskolc-Egyetemváros, Hungary. The research analysis with this software plays an important role in the modelling of the polypropylene product life cycle, using the impact assessment methods [28]. The applied software has provided valuable resources to support consistent modelling of the complete life cycle.

### 2.5. Life Cycle Inventory (LCI) Methodology

The modelling of injection-moulded product systems should use product-specific input data. The accuracy of the life cycle inventory (LCI) method is directly related to the data quality and determines the data objectivity. The applied method includes and quantifies input‒output material flows and energy requirements for all unit processes. This methodology allocates energy requirements and environmental releases among moulded polypropylene products with allocation of mass. The dataset for the polypropylene granules is an annual average. The inventory is mainly based on primary industry data from internationally prevalent production processes.

The resource inputs accounted for include materials and energy use, while the process outputs accounted for include polypropylene products and emissions to the land, air, and water. Using this approach, we examined the injection-moulding process as a looped system by accounting for all resource inputs and process outputs. We used it to allocate environmental burdens for looped polypropylene scrap, looped process water, and polypropylene waste in the production stage. The amount of energy used to heat, cool, and light the injection-moulding space is not included in the system boundaries of this LCI. In the use and end-of-life stages, each dataset includes incoming transportation.

Environmental burdens associated with the end-of-life of the polypropylene products are considered in this research analysis. Cradle-to-grave data for polypropylene products are provided to illustrate the contribution of the converting process to the LCI results for the production of injection-moulded polypropylene products.

The following components of each system are not included in this LCI study: capital equipment, miscellaneous materials, and additives. The applied methodology is consistent with the life cycle inventory methodology described in the ISO 14040:2006 standards [27]. With the help of this approach, we can construct a complete resource and environmental emissions inventory profile for the life cycle of the polypropylene product.

### 2.6. Life Cycle Impact Assessment (LCIA) Method

The life cycle impact assessment (LCIA) phase aims to investigate the possible environmental impacts in the studied system [29]. A wide range of LCIA models were assessed by Legaz et al. [30]. We determined the resources, environmental emissions, and impact categories for all life cycle stages and expressed them in terms of a functional unit of output. A standard unit of output is used as the basis for determining the total life cycle resource requirements and environmental emissions of a polypropylene product of 25 and 1 kg. With the help of the life cycle impact assessment method, we can determine the relative risk to humans or to the environment of emissions from the investigated system.

Among the available midpoint characterization methods in Europe, we applied in this research work the CML (Centrum voor Milieukunde Leiden) method. In the CML method the impact categories have been developed by the Centre for Environmental Science at Leiden University [31,32,33].

During the analysis, the reference system was the total inputs and outputs for the European Union. We used normalization and weighting for calculating the magnitude of the research results of impact category indicators. The applied normalization reference was environmental impacts for 28 European Union countries [27]. The weighting method was thinkstep LCIA Survey 2012 with CML 2016 (excl. biogenic carbon) in Europe. Eight environmental impacts—global warming, acidification, eutrophication, photochemical oxidant formation, human toxicity, abiotic depletion (fossil and elements), and marine aquatic ecotoxicity—were used for the research analysis. Public concern about climate change is an important consideration in modern society, and this environmental impact is measured through the global warming potential. The global warming potential value is for 100 years, excluding. biogenic carbon. The ecosystem protection is related to substances such as sulphur, nitrogen oxides, and phosphorous compounds, which are directly related to the acidification potential and eutrophication potential. The formation of tropospheric ozone threatens the environment and human health and it was considered in the photochemical ozone creation potential. Abiotic resource depletion is one of the most debated impact categories; see, for example, the guidelines of the International Reference Life Cycle Data System (ILCD) and the PEF (Product Environmental Footprint), in Europe [34,35,36]. Human toxicity potential describes the exposure to, and effects of, toxic substances. Guinée and Heijungs [37] decided to base the characterization model of abiotic resource depletion on physical data on reserves and annual deaccumulation. Oers et al., decided that the implementation of substitution options was not (or not yet) feasible within LCA [38]. Marine aquatic ecotoxicity refers to the impact of toxic substances emitted to marine aquatic ecosystems.

By the assessment of the life cycle stages of the polypropylene product and its optimal approaches, the life cycle assessment results of products were compared by quantifying the environmental impacts and emissions. First, the functional unit value was 25 kg of polypropylene product. For further analysis, the value of the functional unit was 1 kg of polypropylene product, where different environmental impacts were examined for the total life cycle of the polypropylene product. The normalization and weighting methods were the same for all analyses performed. The environmental impacts were calculated in all stages for the investigated product and summarized to compare the total impacts of the product. LCIA indicators are described in detail in the Global Guidance on Environmental Life Cycle Impact Assessment Indicators Volume 2 report, which is a follow-up to the Part I report on “Pellston Workshop Environmental LCIA Indicators” published in 2016 [39].

## 3. Results and Discussion

### 3.1. Life Cycle Stages Setup Process

In the research analyses, we examined the total life cycle of a polypropylene product from the raw material extraction and production phase through its usage to the end-of-life stage. The total life cycle of a polypropylene product can be divided into three stages, and numerous factors and loads must be considered. Within the total life cycle, the end-of-life phase plays an important role. The stages of the life cycle of the polypropylene product, based on the environmental product declaration (EPD) technological modules, are as follows [40]:A1‒A3: Production stage: supply of raw materials (polypropylene granules, compressed air, and top water from EU), energy supply (Hungarian electricity grid mix), and injection moulding.A4: Transport of the polypropylene product for use.B1‒B7: Use stage: washing of the polypropylene product, energy use (thermal energy from natural gas and Hungarian electricity grid mix), and deionised water use.C1‒C4: End-of-life stage: transport of used polypropylene product as waste, polypropylene waste disposal in EU-28 waste incineration plant, EU-28 municipal wastewater treatment, credit for steam and power.

First, the mass and energy values of input‒output parameters for the main life cycle stages of the polypropylene product were determined, such as polypropylene granules, compressed air, process water, electricity mix, and thermal energy. Subsequently, the waste streams (wastewater, polypropylene scrap in the production phase, and polypropylene waste in the end-of-life phase) generated in the different stages were determined to see what waste could possibly be returned to the system. We set up the life cycle processes and life cycle plans for each life cycle stage in the LCA software.

### 3.2. Results of Production Stage

The production stage of the polypropylene product is based on an injection-moulding process under actual operating conditions. This involved injection-moulding the polypropylene granules as well as shipping the polypropylene product to be used, which may have led to high environmental impacts for that stage. The raw materials for the injection moulding are polypropylene granules, compressed air, and tap water, where 30 kg of polypropylene granules form 25 kg of polypropylene product in 1 h. The distribution of the raw materials from the extraction sites to the production point was included in the production stage, with relevant transport processes. The polypropylene granules are a polypropylene technology mix (0.86–0.95 g/cm^3^, 43 MJ calorific value) from Germany. At the production stage, we planned for a 20% maximum product loss (based on the recommendations of the Institute of Ceramic and Polymer Engineering at the University of Miskolc).

We examined three production scenarios: (1) without looping method, (2) with process water looping only, and (3) with recirculated plastic scrap looping and process water looping. By using the looping method, we can use dummy processes if we need to connect a flow output into the same process as input. In the second scenario, we planned by the water looping method that 80% of the water flow would be recirculated in a closed loop and 20% of the water flow would be treated as municipal wastewater. Therefore, we can optimize the injection-moulding process with lower environmental loads. The applied life cycle model is completed with municipal wastewater treatment. In the third scenario, the credits for recovered material flows (process water and product loss) were carried out with a looping method. The product loss (5 kg) was recirculated and recycled as plastic scrap into the injection-moulding process. Recycling plastic scraps would reduce the environmental impact, following the cutoff allocation method and neglecting the collection and transport stages.

For the production stage, updated data were extracted from the GaBi 2019 professional database. During the analysis of the injection-moulding process, the background of the raw and auxiliary materials was also considered. These data are representative of the period 2017–2020. The values of life cycle assessment apply to 25 kg polypropylene product in the production stage with transport. The environmental impact associated with the extraction, transport, and refinement of crude oil was also included. The electricity and refinery products are modelled according to the individual country-specific situation. For the transport of polypropylene products, the transport distance was 100 km (utilisation: 61% with diesel mix), considering road transport (Euro 6, 27 t payload, EU-28 diesel mix) in the European Union. The impacts of water consumption on the product systems were assessed with the LCIA methods developed by Pfister et al., as well as the method proposed by Frischknecht et al. [41,42,43].

Figure 1 shows the values of material resources and emissions to freshwater for the three examined scenarios in the production stage. Material resources and emissions to freshwater are larger compared to other flows; therefore, these parameters were illustrated. The percentages of these parameters are 48.56% for material resources and 50.33% for emissions to freshwater. The total value of the other emissions examined (energy resources, deposited goods, emissions to air, emissions to seawater, and emissions to agricultural and industrial soil) was 1.14% for the three scenarios examined. Therefore, these values are negligible and these flows are not shown in Figure 1.

As electricity, we set up a domestic energy mix based on the latest statistical and applied LCA software database. There is no literature available on the Hungarian electricity grid mix yet, which is why we considered the importance of creating and illustrating it here with the help of a pie chart (Figure 2).

Figure 3 shows the relative contribution percentages of input‒output materials and energy flows in the production phase. This pie chart clearly shows that the highest environmental load comes from the polypropylene granule production itself. Table 1 describes the examined environmental impacts in the research analysis.

Figure 4 presents the normalized and weighted values for eight examined environmental impact categories under the actual operating conditions of the injection-moulding process. This figure clearly shows that the marine aquatic ecotoxicity potential and abiotic depletion for fossil fuels are higher compared to other environmental impact categories. The value of abiotic depletion for these elements is negligible.

Our research results agree with studies that found that the total energy requirement for the manufacturing and transport of polypropylene products made these the most energy-intensive stages [44,45]. The examined environmental impacts are higher in the production stage, but these impact categories can be decreased with the investigated and applied looping method. In this production phase, we demonstrated that it may be possible to decrease the impacts on the environment if the polypropylene production process and transport are carried out more sustainably.

### 3.3. Results of Use Stage

In the use stage, we assumed that the polypropylene products are washed. Polypropylene products can be used, for example, in household food packaging. The antecedent of the use stage was the production phase examined by the looping of water and plastic scraps. No specific maintenance requirements are involved in the use stage of the polypropylene product. We modelled a use stage that would be capable of representing a more durable polypropylene product. Reusing the polypropylene product requires a washing process with thermal energy (natural gas from Hungary), Hungarian electricity mix, and deionised water flows. Through the washing process, the introduced 3 kg of deionised water becomes 100% wastewater, but this was released to a wastewater treatment plant in the European Union. So, this use stage was composed of the washing of the polypropylene product and wastewater treatment. During the analysis of the washing process, the background of the raw and auxiliary materials was also considered. The values of the research analysis apply to 25 kg of polypropylene product in the use stage.

Figure 5 shows the relative contribution percentage of input‒output materials and energy flows in the use phase. This pie chart shows that the highest environmental load comes from thermal energy (from natural gas) and an electricity mix. The level of environmental impact caused by wastewater treatment is extremely low.

Figure 6 presents the normalized and weighted values for eight examined environmental impact categories during the use of the polypropylene product. The applied normalization and weighting methods were the same as in the production stage. marine aquatic ecotoxicity potential and abiotic depletion for fossil fuels are higher compared to other environmental impact categories, as in the previous phase. The values of abiotic depletion for elements and eutrophication potential are below 1 ng.

### 3.4. Results of End-of-Life Stage

The life cycle assessment can play an important role in the end-of-life stage. There are some treatment approaches for decreasing the quantity of polypropylene waste, but currently incineration and landfill are the most popular alternatives. Figure 7 presents different solutions for plastic waste management in the end-of-life stage.

Many studies summarise new information for the treatment processes of polypropylene waste with a comparison between the different technologies [46,47,48]. The thermic treatment process (pyrolysis, incineration, gasification, and plasma-based technology) can be considered on the basis of the environmental burden and energy efficiency [49,50,51]. The new technologies enable greater market penetration, since these secondary energy sources are compatible with gas turbines and gas motors [52,53].

The end-of-life stage of the polypropylene product as plastic waste was modelled with waste incineration in a municipal waste incineration plant in the European Union. For the transport of polypropylene waste, the transport distance was 100 km, where we took transport by road, boat, and railway within the European Union into account. The raw materials and energy streams used determine the energy consumption and the environmental impacts of the production, use, and EoL phases, so they can significantly influence the production stage and the total life cycle of the product [54]. The residual steam and electric power should be reused in a specifically designed plant [55]. In the end-of-life stage, the waste treatment was solved by conventional incineration, where 35 MJ of heat energy and 66 MJ of electric power were generated automatically by the GaBi software from 25 kg of plastic waste. In this stage, we can calculate with energy feedbacks. Credits for steam and electric power can be substituted for the primary energy supply (energy recirculation). In the long term, the most economical technologies are environmentally friendly and energy-saving ones that can raise the economic efficiency and innovation index of a company [56,57].

Figure 8 presents the normalized and weighted values for eight examined environmental impact categories in the end-of-life stage of the tested product. The applied normalization and weighting methods were the same as in the production and use stage.

According to Figure 8, we can determine that the global warming potential for 100 years is higher compared to other environmental impact categories. Abiotic depletion for elements and eutrophication potential are negligible compared to the other environmental impact categories.

### 3.5. Results for the Complete Life Cycle of a Polypropylene Product

Table 2 shows the absolute values of energy and material resources, and emissions associated with the total life cycle for a functional unit of 1 kg polypropylene product. Here, the production stage was considered without a looping method. The highest resources and emissions of the product were perceived in the production stage. If the relative contribution values of these flows during the total life cycle for the polypropylene product are considered to be 100%, then the following results are obtained by examining the individual stages: 97% in the production stage, 0.89% in the use stage, and 2.5% in the end-of-life stage. According to Table 2, we can determine that material resources and emissions to freshwater show outstanding values in the three life cycle stages. These relative contribution values are 48% and 50% for the total life cycle of the polypropylene product. Emissions to agricultural and industrial soil are negligible as these values are almost zero.

Table 3 shows the absolute values of resources and emissions associated with the total life cycle of a polypropylene product with looping of process water and plastic scrap. The highest resources and emissions for the product were observed in the production stage. The relative contribution was 96% in the production stage, which means that the applied looping method reduces the total resources and emissions by 1%. According to Table 3, we can determine that the largest change is observed in air emissions. The looping method, used in the production stage, can reduce resources and emission values by 0.93% over the total life cycle of the tested product.

Table 4 presents the normalized and weighted values for 10 different environmental impact categories associated with the total life cycle of 1 kg of polypropylene product. A looping method was used in the production stage. The value of ozone layer depletion potential is zero in all stages. Therefore, this environmental impact category is not listed in Table 4.

Table 5 shows the values of eight environmental impact categories associated with the total life cycle of 1 kg of polypropylene product. The impact categories (ADPE, EP, and ODP), which comprise less than 1% of the system mass, have a negligible effect on the total life cycle assessment results. These environmental impacts are not listed in Table 5.

Figure 9 shows the quantity view for all environmental impact categories under actual operating conditions from the injection-moulding process of 30 kg of polypropylene granules using a 25 kg polypropylene product until it becomes waste. The effect category values shown in the figure for the production, use, and end-of-life stages were determined for the actual operating conditions.

If the total value of the environmental impacts during the whole life cycle of the product is considered to be 100% (relative contribution, the sum of environmental impact categories examined by the LCA software as a percentage), it is distributed as follows: production phase 91%, use phase 3%, and end-of-life phase 6% burden on the environment. The total life cycle of the polypropylene product mainly causes a larger change in the relative contribution values of the environmental impact categories MAETP (Marine Aquatic Ecotoxicity Potential, 44%) and ADPF (Abiotic Depletion for fossil fuels, 28%). In the production stage, the values for ADPF and MAETP are dominant at 27% and 43%, respectively.

In the use stage, we get the highest percentage for ADP fossil (1.0%) and the lowest for ODP (0.35%). In the end-of-life stage, we get the highest percentage value for ADPF (1%) and the lowest value for TETP (Terrestrial Ecotoxicity Potential). This value is 0.0024%. The life cycle of the polypropylene product affects AP (Acidification Potential), HTP (Human Toxicity Potential), and POCP (Photochemical Ozone Creation Potential) values to a small extent (3–6%) and has a minimal effect on the FAETP (Freshwater Aquatic Ecotoxicity Potential), ADPE (Abiotic Depletion for elements), and EP (Eutrophication Potential) values (these are below 1.3%).

If the values of the environmental impacts occurring during the total life cycle of the product are considered collectively to be 100% (in relative percentage), then the values for each impact category total between 60% and 99% in the production stage. In terms of global warming for 100 years with region equivalents weighted excl. biogenic carbon, the percentage is the highest (60% relative percent) in the production stage. Values for ODP (Ozone Layer Depletion Potential), MAETP, and FAETP are very high in this stage (97–99%).

## 4. Conclusions

This study examined the total life cycle of a polypropylene product, from the raw material extraction and injection moulding through its usage to the end-of-life, and attempted to determine its impact on the environment. The analysis lasted from the cradle to the grave, with scenario analysis of the looping method in the production phase. The applied life cycle model is completed with municipal wastewater treatments and waste incineration. By quantifying the environmental impacts of all life cycle stages, the CML analysis method was applied by GaBi 8.0 thinkstep software. The normalization and weighting methods were the same for all life cycle stages.

The functional unit was defined as the distribution of 25 kg of polypropylene product output for all stages. The functional unit was redefined as 1 kg of product output. The highest environmental effects of the product were recorded in the production stage. In this phase, the recycling of material flows (cooling water, polypropylene loss) was carried out by the looping method. Material and energy input-output flows of the injection moulding have a significant impact on the life cycle of the polypropylene product.

According to the results of the life cycle assessment, the emissions and environmental impacts in the end-of-life stage are smaller compared to the production stage but higher compared to the use stage. If the total value of the environmental impacts during the whole life cycle of the product is considered to be 100% (relative contribution, the sum of environmental impact categories examined by the LCA software as a percentage), it is distributed as follows: production phase 91%, use phase 3%, and end-of-life phase 6% burden on the environment. The rate of global warming potential (GWP 100 years excluding biogenic carbon) over the total life cycle of the product for 100 years is 60%, 5%, and 35%, respectively, for the production, use, and end-of-life stages.

To make a complete decision on the environmental friendliness nature of a technological process is a complex task. Basically, for sustainable production it is essential to evaluate the complete life cycle of the polypropylene products. There are very poor professional literatures with complete lifecycle research topic for plastic products. We did not find any studies that look at the complete life cycle model with the looping method for polypropylene products.

The research results can be used to develop the injection-moulding process with lower environmental impacts and these can be applied to further research on the manufacturing processes of other polypropylene products. The results can be used by companies to support product-orientated environmental management by users of plastic products as a building block of life cycle assessment studies of individual products, and by other interested parties, as a source of life cycle information.

## Figures and Tables

**Figure 1 polymers-12-01901-f001:**
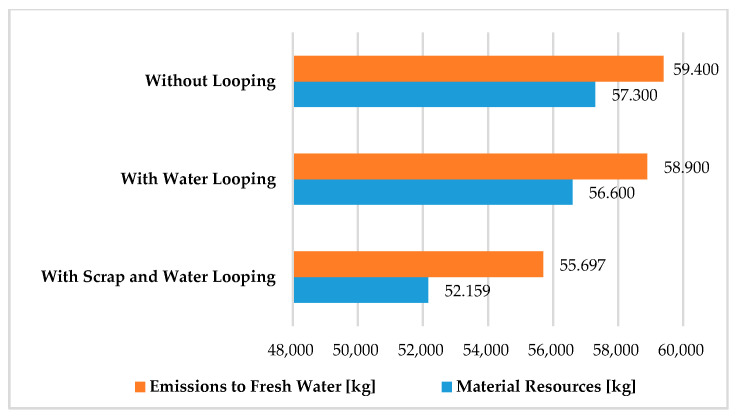
Values of material resources and emissions to fresh water in the production stage for the examined production solutions with transport. (Functional unit: 25 kg polypropylene product. Normalization unit: kilogram, weighting quantity: mass).

**Figure 2 polymers-12-01901-f002:**
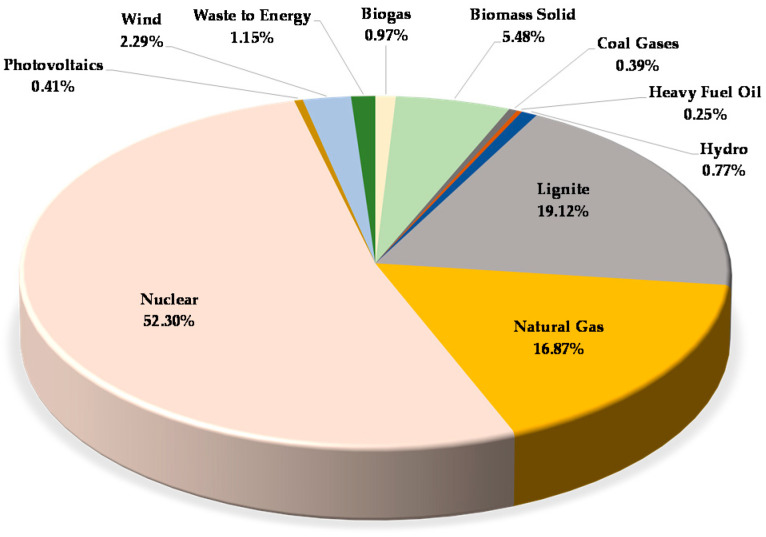
Hungarian electricity mix.

**Figure 3 polymers-12-01901-f003:**
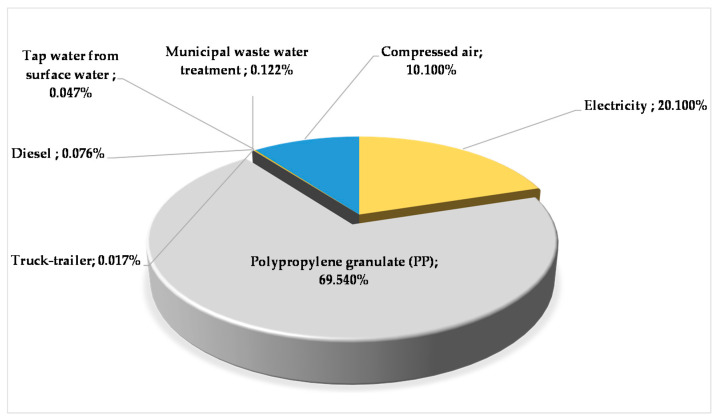
Relative contribution percentage of input‒output materials and energy flows in the production stage.

**Figure 4 polymers-12-01901-f004:**
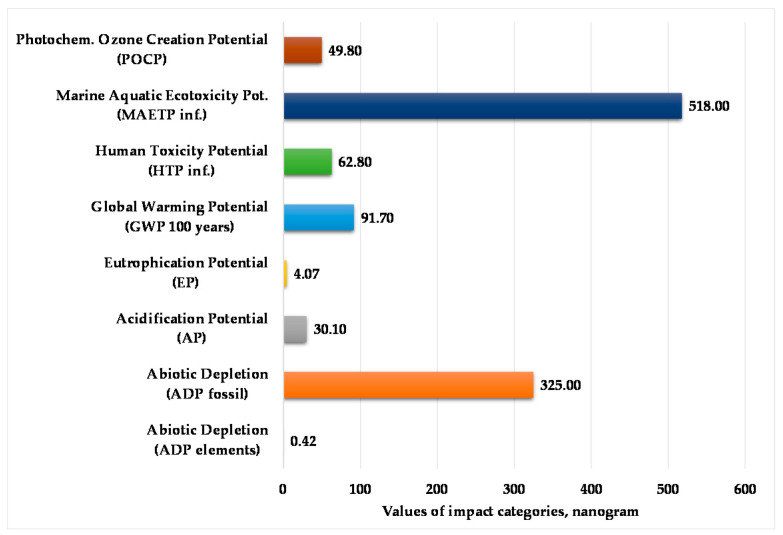
Values of environmental impacts in the manufacturing phase with transport. (Functional unit: 25 kg polypropylene product. Normalization reference: CML 2016 (Centrum voor Milieukunde Leiden, 2016), EU 25 + 3, year 2000, excl. biogenic carbon. Weighting method: thinkstep life cycle impact assessment (LCIA) Survey 2012, Europe, CML 2016, excl. biogenic carbon).

**Figure 5 polymers-12-01901-f005:**
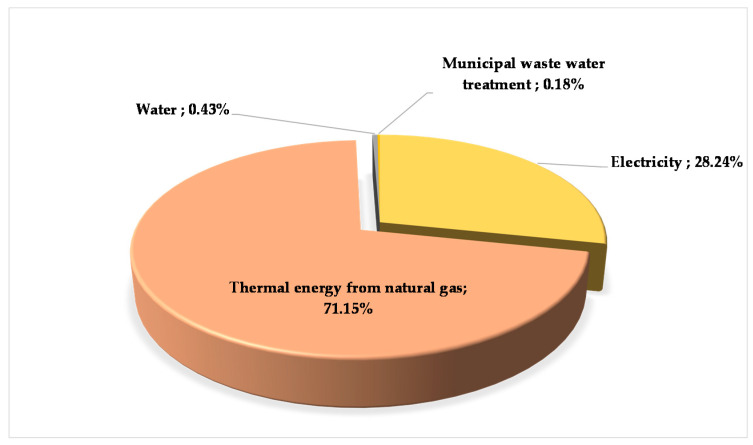
Relative contribution percentage of material and energy flows in the use phase.

**Figure 6 polymers-12-01901-f006:**
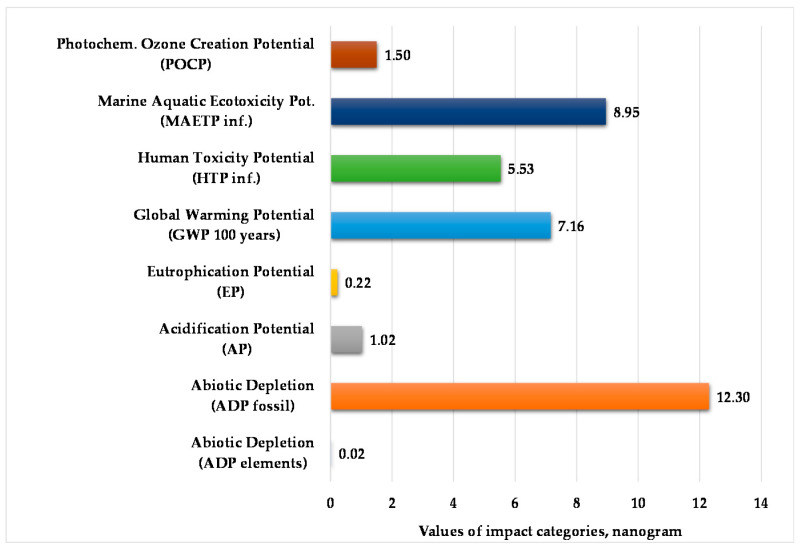
Values of environmental impacts in the use phase with transport. (Functional unit: 25 kg polypropylene product. Normalization reference: CML 2016, EU 25 + 3, year 2000, excl. biogenic carbon. Weighting method: thinkstep LCIA Survey 2012, Europe, CML 2016, excl. biogenic carbon).

**Figure 7 polymers-12-01901-f007:**
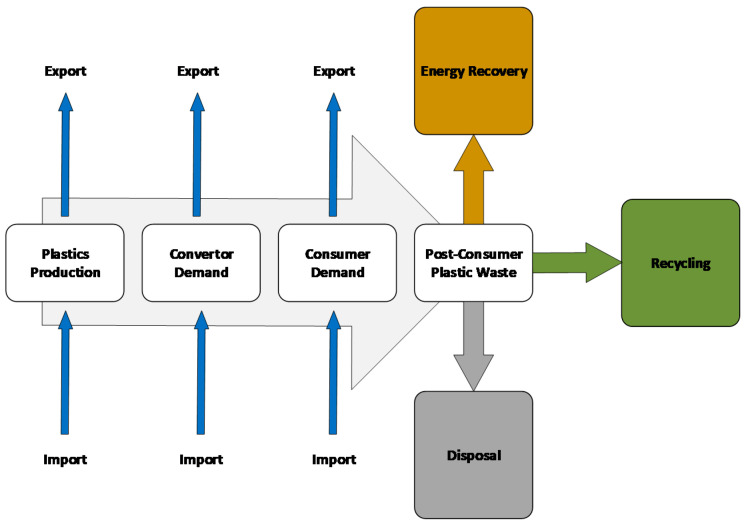
Solutions for the polypropylene waste management in the end-of-life stage.

**Figure 8 polymers-12-01901-f008:**
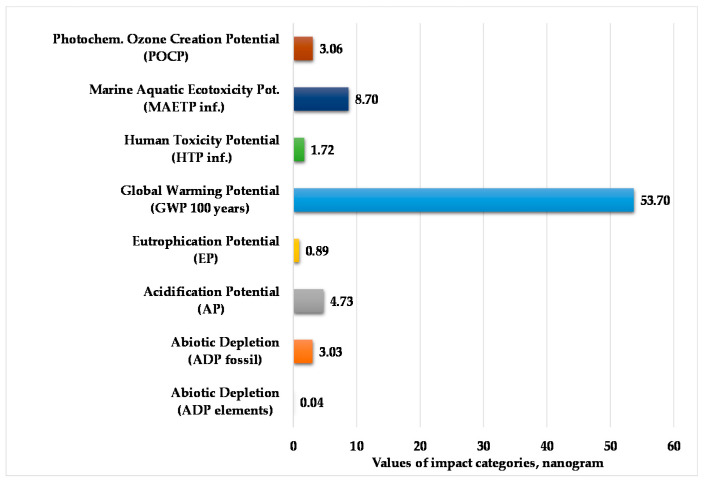
Values of environmental impacts in the end-of-life (EoL) phase. (Functional unit: 25 kg polypropylene product. Normalization reference: CML 2016, EU 25 + 3, year 2000, excl. biogenic carbon. Weighting method: Thinkstep LCIA Survey 2012, Europe, CML 2016, excl. biogenic carbon).

**Figure 9 polymers-12-01901-f009:**
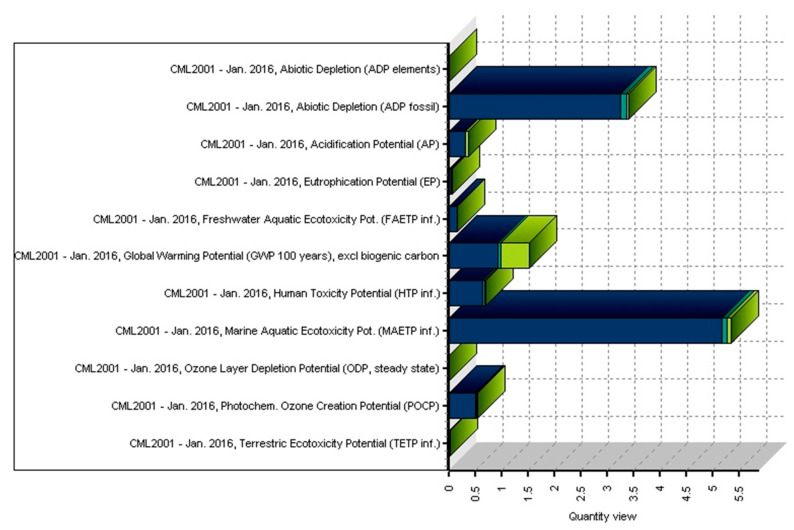
Quantity view for all examined impact categories under actual operating conditions (functional unit: 25 kg polypropylene product. Colours: Dark blue—production phase, light blue—use phase, green—EoL phase).

**Table 1 polymers-12-01901-t001:** The examined environmental impact categories [28,33].

Impact Categories	Equivalent
Abiotic Depletion *ADP elements*, *ADPE*	kg Sb Equivalent
Abiotic Depletion *ADP fossil, ADPF*	MJ
Acidification Potential *AP*	kg SO_2_ Equivalent
Eutrophication Potential *EP*	kg Phosphate Equivalent
Freshwater A. Ecot. P. *FAETP inf.*	kg DCB Equivalent
Global Warming Pot. *GWP 100 years*	kg CO_2_ Equivalent
Human Toxicity Potential *HTP inf.*	kg DCB Equivalent
Marine A. Ecotox. Pot. *MAETP inf.*	kg DCB Equivalent
Photochem. Ozone Creat. Pot. *POCP*	kg Ethylene Equivalent
Terrestric Ecotox. Pot. *TETP inf.*	kg DCB Equivalent
Ozone Depletion Pot. *ODP steady state*	kg R11 Equivalent

**Table 2 polymers-12-01901-t002:** Absolute values of resources and emissions in the life cycle stages without looping. (Functional unit: 1 kg of polypropylene product. Weighting quantity: mass, normalization unit: kilogram. Impact assessment method: CML 2016).

Flows	Prod.	Use	EoL	Total LC
Energy resources	2.01	0.06	0.02	2.09
Material resources	2290.00	19.20	59.30	2369
Deposited goods	4.37	0.13	0.27	4.77
Emissions to air	46.00	1.37	9.64	57.01
Emissions to freshwater	2370.00	22.70	52.50	2445.00
Emissions to seawater	7.20	0.01	0.18	7.39
**Flows**	**4720.00**	**43.47**	**122.00**	**4885.00**

**Table 3 polymers-12-01901-t003:** Absolute values of resources and emissions in the life cycle stages with looping. (Functional unit: 1 kg of polypropylene product. Weighting quantity: Mass, normalization unit: kilogram. Impact assessment method: CML 2016).

Flows	Prod.	Use	EoL	Total LC
Energy resources	1.89	0.06	0.02	1.97
Material resources	2090.00	19.20	59.30	2168.50
Deposited goods	5.1	0.13	0.27	5.50
Emissions to air	38.70	1.37	9.64	49.71
Emissions to freshwater	2230.00	22.70	52.50	2305.20
Emissions to seawater	6.54	0.01	0.18	6.73
**Flows**	**4372.23**	**43.47**	**121.91**	**4537.61**

**Table 4 polymers-12-01901-t004:** Results of total life cycle of the product in the different life cycle stages (with looping). (Functional unit: 1 kg of polypropylene product. Normalization: CML 2016, EU 25 + 3, year 2000, excl. biogenic carbon. Weighting method: thinkstep LCIA Survey 2012, Europe, CML 2016, excl. biogenic carbon).

Impact Categories	Prod.	Use	End-of-Life
Abiotic Depletion *ADP elements*, ng	0.017	0.001	0.002
Abiotic Depletion *ADP fossil*, ng	13.000	0.494	0.121
Acidification Potential *AP*, ng	1.200	0.041	0.189
Eutrophication Potential *EP*, ng	0.163	0.009	0.036
Freshwater A. Ecot. P. *FAETP inf.*, ng	0.615	0.005	0.004
Global Warming Pot. *GWP 100 years*, ng	3.670	0.286	2.150
Human Toxicity Potential *HTP inf.*, ng	2.510	0.221	0.069
Marine A. Ecotox. Pot. *MAETP inf.*, ng	20.700	0.358	0.348
Photochem. Ozone Creat. Pot. *POCP*, ng	1.990	0.060	0.123
Terrestric Ecotox. Pot. *TETP inf.*, ng	0.065	0.001	0.020
**Total value of environmental loads**	**44.000**	**1.480**	**3.060**

**Table 5 polymers-12-01901-t005:** Environmental impacts in the life cycle stages of the polypropylene product (FU: 1 kg of polypropylene product. Impact assessment method: CML 2016).

Impact Categories	Prod.	Use	End-of-Life
Abiotic Depletion *ADP fossil*, MJ	71.300	2.710	0.664
Acidification Potential *AP*, kg SO_2_ eq	0.004	0.000	0.001
Freshwater A. Ecot. P. *FAETP inf.*, kg DCB eq.	0.021	0.000	0.000
Global Warming Pot. *GWP 100 years*, kg CO_2_ eq	2.180	0.170	1.270
Human Toxicity Potential *HTP inf.*, kg DCB eq.	0.193	0.017	0.005
Marine A. Ecotox. Pot. *MAETP inf.*, kg DCB eq.	149.000	2.580	2.510
Photochem. Ozone Creat. Pot. *POCP*, ng	0.001	0.000	0.000
Terrestrial Ecotox. Pot. *TETP inf.*, kg DCB eq.	0.001	0.000	0.000

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
