# Peer review of "Total Life Cycle of Polypropylene Products: Reducing Environmental Impacts in the Manufacturing Phase"

_polymers, 2020, doi:10.3390/polym12091901_

Round 1

Reviewer 1 Report

In my opinion this paper is interesting and after minor revision can be considered for publication.

Please check the values in Figure 5 (sum of all values is not equal 100%).

My detailed comments in the attachment.

Author Response

Dear Honourable Reviewer,

we would like to kindly thank you again for your thorough assessment of our article, your time and your patience. We highlighted the changes in yellow in our article. Our detailed responses in the pdf attachment.

Yours Faithfully,

Viktoria Mannheim and Zoltan Simenfalvi

authors

Reviewer 2 Report

Dear Authors,

The paper was seriously improved after revision.

Best regards,

Author Response

Dear Honourable Reviewer,

we would like to kindly thank you again for your thorough assessment of our article, your time and your patience. We accepted with gratitude all the valuable comments of the Honourable Reviewer.

Yours Faithfully,
Viktoria Mannheim and Zoltan Simenfalvi
authors

This manuscript is a resubmission of an earlier submission. The following is a list of the peer review reports and author responses from that submission.

Round 1

Reviewer 1 Report

Dear authors,

This paper presents the LCA results of the total life cycle of polypropylene products. The paper did not provide an overview of Life Cycle Assessment in terms of the holistic environmental impact of the production and delivery of a product, covering carbon emissions, wastewater, solid waste, by-products, and so on. 

The paper could be published after major revision concerning the following aspects:

  • establish the objectives of the research
  • discuss the goal and scope of the assessment
  • describe the life cycle inventory modelling
  • the results are nor properly interpreted

Best regards,

Reviewer 2 Report

In my opinion this paper is not suitable for publication in Polymers journal.

Current, version of the manuscript is rather technical raport than scientific paper. There is a lack of discussion with comparision of obtained results with other research works. 

Moreover, the Authors have tendency to use many abbreviations without prior explanaion in text, which sometimes is very confusing.

My detailed comments in the attachment.

Reviewer 3 Report

The reviewer is divided regarding the evaluation of the present manuscript. On the one hand, it reports a well done Life Cycle Assessment study, denoting that the authors know all the requisites of this scientific domain. It also presents clear and relevant research questions (not all equally well addressed later in the text). But, on the other hand, it is rather badly written, in some sections the English being so appalling that the conceptual aspects are incomprehensible. Some of the figures are quite difficult to understand and need to be redrawn and better explained (as indicated in the specific comments). Other figures appear in the wrong page and are only mentioned afterwards in the text. There are possibly too many references, particularly from Hungarian authors, and others more general and equally (or, perhaps, more) important are missing, e.g. some relevant EU Directives. There are also many acronyms presented without prior (or concomitant) explanation; an acronyms glossary is therefore mandatory. There are also some conceptual contradictions in the text that must be corrected.

Summing up: balancing its positive and negative aspects, the manuscript may be accepted after a proper revision that clearly must be more stringent than minor.

Specific comments:

Page 1. Keywords: “Life Cycle Assessment (LCA)” is missing!

Introduction lines 27-37:  the text on polypropylene (PP) - and plastics, in general - is trivial, unnecessary for a polymer specialist and irrelevant for a LCA practitioner;

The sentence on injection moulding processes (line 38) is ridiculous and wrong; just consider the importance of blow moulding and extrusion-based processes for packaging and that of packages in terms of environmental impact.

Page 2: it is unconceivable why references [2] and [3] are presented and not the Waste Framework Directive (Directive 2008/98/EC of 19 November, Official Journal of the European Union, L 312/3-30); also references [3] to [7] are too many to substantiate the assertion; they could be substituted advantageously by a more targeted one: Plastics-the Facts 2019 from PlasticsEurope.

The assertion on innovative injection moulding - IM - technology (line 70) is an overstatement; actually IM is nowadays a very mature and stable technology with just incremental innovation.

Page 3 line 90: There are no losses in the IM process?  Not likely, even considering reprocessing!

Page 4 lines 124/125: References other than [25, 27] would certainly be advisable.

Page 5 lines 161/162: reprocessing cannot guarantee that there are no material losses in the process! Furthermore, exactly how was the figure 20% scrap flow obtained?

In Figure 2 it is not clear what do the ordinates mean, particularly 52159 & 55697 kg? To what weight of PP do they refer to?

Page 6 Figure 3: the data therein seems OK, but, with the explanations provided, only LCA practitioners will understand it.

In Figure 4, the contributions in the pie chart add up to 100.1%.

Page 7 line 187: the assertion “Usually plastic products are used once and disposed” is a senseless overgeneralization; most technical plastic parts are used continuously, at least, as often as those made from other materials; even some packages (personal plastic bottles, for instance) are nowadays multiply reused.

Figure 5:  again it is not clear what do the ordinates mean; to what weight of PP do they refer to?

In lines 206/207 “Figure 4” and “manufacturing” should be “Figure 6” and “use phase”

Page 8 line 213: “dissipative use”!!!

Page 9 Figure 8: once more, it is not clear what do the ordinates mean; to what weight of PP do they refer to?

Page 10 Figure 10: the contributions in the pie chart seem incomplete; what about the heat and electricity flows mentioned in the text?

Page 11 Table 2: it is not clear how the total values were obtained; values in MJ and kg cannot be summed!!

Figure 11 is confusing: most impacts are not represented or have zero values - why mention them?

Pages 11/12: the reviewer could not conciliate the data on Figures 11 and 12; their relation must be better explained.

Page 13 Figure 14: the caption mentions yellow to represent the use phase but no yellow is visible; possibly it should be light blue.

Pages 13/14: the discussion is probably the section of the manuscript where the grammar is worst; it is almost impossible to understand the concepts therein.

A final opinion: the figures are very “cartoon-like”, specially n. 2, 3, 5, 6, 8 and 9; it would seem better to “normalize” them in the revised version.